# Validity of the German version of the Stay Independent Questionnaire applied by telephone interview: A diagnostic accuracy study

Ulrich Thiem[1,2]*, Ingeborg Schlumbohm[3], Stefan Golgert[3], Ulrike Dapp[3], Saskia Otte[3]

1 Department of Geriatrics, Marienhospital Herne, University Clinic, Ruhr-Universität, Bochum, Germany, 2 Department of Geriatrics, Medical School OWL, University of Bielefeld, Bielefeld, Germany, 3 Department of Geriatrics, Albertinen-Haus, Hamburg, Germany

* ulrich.thiem@ruhr-uni-bochum.de

## Abstract

### Background

Mobility limitations are among the most common functional problems in older people. Repeated falls can lead to injuries and fractures, trigger or intensify concerns of falling, and contribute to subsequent functional decline and loss of independence. Various questionnaires have been developed, both nationally and internationally, to identify older people at increased risk of falling. Data for evaluation against standard tests from the geriatric mobility assessment are scarce.

### Methods

In a German project evaluating home emergency call systems, the Stay Independent Questionnaire (SIQ) from the American prevention program STEADI (Stopping Elderly Accidents, Deaths, and Injuries) was used for the identification of community-dwelling seniors aged 70 and older at risk of falling. The original questionnaire was translated by professional translators using the typical forward and backward translation process, and a final version was established after discussion involving a bilingual scientist. The diagnostic performance of the questionnaire (diagnostic test) was tested against the mobility assessment Short Physical Performance Battery (SPPB, gold standard). To describe the test performance, typical statistical measures are used, i.e., sensitivity, specificity, positive and negative predictive values, and positive and negative likelihood ratios, each with the corresponding 95% confidence interval (95% CI).

### Results

Data from a total of 190 participants (143 women, 75.3%; average age 80.5 years±5.5 years standard deviation) were included in the analysis. According to

**Data availability statement:** Data from this study are not publicly available due to legal issues, especially German data protection regulations. The Ethics Committee of the Hamburg Medical Association did only allow data collection, processing and storage for this particular project and research question. Furthermore, participants have not been informed about a secondary use of data by other researchers, and hence did not give consent to this. A further use of our data would demand a new application at the Ethics Committee of the Hamburg Medical Association. The Albertinen Office for Data Protection (contact: datenschutz@immanuelalbertinen.de, phone: +49 151 70645743) will be ready to assist researchers with interest in using the data.

**Funding:** This validation study as well as the RCT evaluating home emergency call systems is funded by the German Innovation Fund of the Federal Joint Committee (FKZ 01NVF21102) awarded to UT and UD. The sponsor played no role in the design or conduct of the study, the analysis or interpretation of data, the drafting of the manuscript or the decision to publish it.

**Competing interests:** The authors have declared that no competing interests exist.

existing comorbidities and functional abilities, between 30% and 40% suffered of advanced comorbidity and/or functional impairment. The questionnaire identified 148 individuals (77.9%) as at risk of falling. According to SPPB, 81 participants had an objectively measurable impairment of standing and walking balance. The test performance measures for SIQ as a diagnostic test are: sensitivity 95.1% 95% CI [88.0%; 98.1%], specificity 34.9% [26.6%; 44.2%], positive and negative predictive value 52.0% [44.0%; 59.9%] and 90.5% [77.9%; 96.2%], respectively, and positive and negative likelihood ratio 1.46 [1.26; 1.69] and 0.14 [0.05; 0.38]. In receiver-operating characteristic (ROC) analysis, the unadjusted area under the curve for SIQ was 65.0% [57.3%; 72.7%], after adjustment for sex and age 71.0% [63.8%; 78.2%].

## Conclusions

The German version of the Stay Independent Questionnaire is capable of identifying community-dwelling seniors aged 70 and older without impairment in standing and walking balance. The high sensitivity of the test allows excluding test-negative individuals from further investigation. A limitation of the questionnaire is the high proportion of false positives, resulting from the low specificity of the test. Scientific evaluation will show to what extent the use of the questionnaire may improve the identification and medical care of community-dwelling seniors at risk of falling in terms of fall prevention.

## Background

Falls, as an expression of mobility limitations, are a common and serious functional problem in older individuals and geriatric patients. It is estimated that at least one fifth of people aged over 65 years fall at least once a year [1]. The frequency varies in different populations and is highest among the very old and individuals in nursing care facilities [1,2]. Repeated falls can lead to a diminished quality of life due to pain and injuries [3,4]. They can trigger or worsen concerns of falling and may contribute to further functional decline and loss of independence [5]. Feared consequences of falls include fractures, particularly proximal femoral fractures, but also fractures and injuries of other locations. In Germany, falls directly cause about 10,000 deaths annually [6]. Among adults, higher age groups, especially those aged 80 years and older, and women in particular are affected [6,7]. The numbers are increasing [6], as is the incidence of fractures associated with falls [8]. Given the aging population, falls and fall related injury are becoming an increasing challenge.

The burden on the healthcare system due to costs associated with falls is considerable. In the USA, healthcare costs for managing fatal fall-related injuries were over $600 million, and costs for non-fatal fall injuries exceeded $30 billion for the years 2012 and 2015. Between 2012 and 2015, an increase of expenditures of about 3.5% was noted, with a projected upward trend in subsequent years due to increasing fall rates, especially among older age groups [9,10]. Comparable health economic

consequences and trends are reported for China [11]. One reason for the financial burden is the repeated utilization of medical services by individuals after one or more fall events [12].

As a low-threshold approach to identifying community-dwelling seniors at increased risk of falling, various self-report questionnaires and models have been developed both nationally and internationally. Examples are: Geriatric Postal Screening Survey (GPSS, Alessi, 2003) [13]; Fall Risk Check (Sturzrisiko-Check, SRC, Anders, 2006) [14]; Falls Risk of Older People in the Community (FROP-COM, Russell, 2008) [15]; Falls Risk Questionnaire (FRQ, Rubenstein, 2011) [16]; models by Hirase [17]and Gadkaree [18]; and the online questionnaire by Obrist [19]. The mentioned instruments vary considerably with regard to included items, length and ease of use. The Fall Risk Check, one of two German questionnaires, covers 13 areas, each with between two and seven individual statements – 50 in sum – to be checked and made [14,20]. The online questionnaire by Obrist, the other German-language tool, developed in Switzerland, consists of 29 predictors of fall risk with a total of 36 questions [19]. A very short model by Gadkaree uses only five items – age, sex, ethnicity, and falls and balance problems by self-report. However, the test performance of this model is rather limited [18]. For most tools, evaluation has been limited, with often only one single publication describing the development of the instrument.

The prevention program STEADI (STopping Elderly Accidents, Deaths, and Injury) initiated by the American Center of Disease Control and Prevention (CDC) [21] uses a variant of the Fall Risk Questionnaire (FRQ) developed by Rubenstein [16], called Stay Independent Questionnaire (SIQ). The questionnaire includes twelve simple yes/ no questions that cover areas such as the number of falls in the past year, concerns of falling, sensory impairments, dizziness, medications that may increase falls risk, and tendency towards depression. Although the questionnaire itself was not extensively validated, several studies evaluated the STEADI program as a whole and rate it positively [22–25]. According to a recently published implementation study, the STEADI program is effective in preventing falls and their consequences, and is also cost-effective [26].

For the identification of older individuals with increased risk of falls, the World Guideline for Falls Prevention recommends asking a single question about a fall within the past twelve months [1]. According to a meta-analysis, the single question has assumed to be highly sensitive to fall risk [27]. Unfortunately, the sensitivity estimate appears to be biased, as it has been derived from longitudinal studies with continuous falls monitoring by periodical enquiry, thus improving the recall of study participants. Another shortcoming of the single question is that individuals with an increased risk, but without fall will not be recognised. Therefore, more detailed questionnaires are needed that can be administered quickly, easily, and at low cost.

For the recruitment of community-dwelling seniors with increased falls risk for a randomised controlled trial (RCT) investigating home emergency call systems, we needed an easy to use questionnaire suitable for use in a telephone interview. We chose the SIQ, as it is shorter and easier to use than the known German questionniares, and decided to translate it into German and validate it against a standard mobility assessment.

## Methods

### a) Study hypothesis

Our hypothesis was that the translated version of SIQ is able to validly identify community-dwelling seniors aged 70 and over without impairment in standing and walking balance. Sufficient validity was assumed for SIQ as a diagnostic test with a sensitivity of at least 75% or higher. For diagnostic tests, a high sensitivity is required to exclude unimpaired (healthy) individuals [28].

### b) Recruitment of study participants

Participants for this diagnostic study were recruited as part of the recruitment for the main study, a German RCT investigating different home emergency call systems (INES, Intelligentes NotfallErkennungsSystem). Details of the main study have been published recently [29]. In brief, participating health insurance companies contacted potentially eligible insured persons in three federal states in Germany – Hamburg, Bavaria, North Rhine Westphalia – and invited them to participate

in the RCT. Criteria for contacting included age ≥ 70 years, presumably living alone, and written consent for further contact. Initially, a selection based on ICD-10 codes (International Classification of Diseases, 10th edition, German Modification) [30] coded in the insurers documents was made to increase the number of individuals with diseases increasing falls risk. The ICD codes in use were: G2x – Parkinson's syndromes; G8x – paresis; H8x – vestibular function disorders/ vertigo; I63.x – stroke; R26.x – mobility disorders; R29.6 – fall tendency, U5x – functional impairments. Recruitment began at July 17th, 2023. From mid-October 2023 (10/19/2023), insured persons irrespective of specified ICD codes were contacted and invited to participate. Recruitment ended at March 7th, 2024, with the appointment of the last participant.

Consenting individuals were called by a call centre and screened for eligibility. The main criterion for eligibility was an increased fall risk identified by SIQ. Individuals at risk were invited to participate in the RCT, and relevant data were transferred to cooperating providers of home emergency call systems for further procedures.

Regardless of the SIQ results, insured persons residing in Hamburg and contacted by the call centre were additionally offered participation in this sub-study. Thus, insured individuals aged ≥ 70 years residing in Hamburg and living alone were included in the SIQ validation study, regardless of their fall risk assessment using SIQ. For a period of five weeks, from early August to mid-September 2023, recruitment by the call centre had to be suspended to handle the initially large number of people willing to participate. Otherwise, all consenting individuals were recruited consecutively. No compensation was provided for participation in the study. Since potentially mobility-impaired individuals were to be included and examined, reimbursement of travel expenses was offered. About a third of the participants claimed reimbursement for travel or taxi costs. As an additional incentive, a written summary and evaluation of the findings was sent to the participants, if desired.

The study personnel who conducted data collection for this study remained blinded to the data collection of the call centre and, in particular, to the information from SIQ. Due to the sequence of recruitment, a randomised administration of SIQ and SPPB was not possible.

### c) German translation of the stay independent questionnaire

The original version of the questionnaire was developed and published by L. Rubenstein et al. in 2011 [16]. Until study planning, no validation against subsequent, incident falls within a cohort study was reported, nor testing against a gold standard, e.g., from geriatric assessment. The CDC incorporated the questionnaire into the STEADI prevention program in 2013 [21,31]. Since then, it has been recommended as a screening tool for assessing fall risk in seniors in the USA.

The CDC's questionnaire version was translated into German by one and back-translated by another professional translator, with the second translator being blinded to the original version. The back-translation and the original version were compared, and the translation was finalised by the study group with the help of a bilingual scientist, experienced in geriatric medicine. The German questionnaire was then pre-tested with eight seniors visiting the geriatric day clinic associated to the Department of Geriatrics. After short instruction by staff personnel, the patients voluntarily answered the self-administered German SIQ version. After completion, issues like clarity of the questionnaire, of single items, wording etc. were discussed with the volunteers. Feedback was received, but did not lead to further alterations of the questionnaire.

The call centre used the translated version of SIQ for use in the telephone interviews. Call centre staff were trained in its application during an on-site visit by study personnel from Hamburg. The background and objective of this study were presented, special features of the questionnaire discussed, and potential problems in application debated. During recruitment, study personnel were available to the call centre for questions and discussions.

### d) Test administration

The German translation of SIQ was used by the call centre to determine potential fall risk in potentially eligible seniors. The CDC's evaluation scheme for SIQ was used, which classifies all individuals as at risk if they answer four or more of the twelve questions in SIQ positively, i.e., indicating a fall risk (SIQ total score ≥ 4 points) [32]. Regardless of the total score, all respondents reporting a fall event during the past year are classified as at risk. This corresponds to a weighting

of the fall question (yes = 4 pts.) compared to the other eleven questions of SIQ [32]. The application of the questionnaire at the call centre occurred without knowledge of further clinical information about the participants. As the validation study took place later, the call centre had no participant information from the subproject.

While only fall-risk seniors were considered for recruitment for the RCT, both individuals with and without fall risk according to SIQ were recruited for this validation study. Because of the health insurance companies' eligibility criteria, the call centre recruited more seniors with fall risk than without when contacting potential participants for the main study.

As a gold standard for verifying fall risk, the test battery Short Physical Performance Battery (SPPB) from geriatric mobility assessment was chosen. This test examines standing and walking balance in capable individuals in three different parts. In the standing part, the ability for parallel, semi-tandem, and tandem stand for 10 seconds each is tested. In the walking part, walking speed over a distance of four meters is measured. The third part is the performance of the modified Chair Rise Test (CRT), where the time is measured that participants need to rise up from a chair seat five times without using arms or hands for support (5-CRT). Participants were considered test-positive (i.e., pathological) if they scored fewer than ten points (out of a possible twelve points) on the SPPB total score [33].

Apart from few exceptions, the mobility assessment was conducted by a single female physiotherapist, experienced in geriatric medicine. In cases of questions about the execution, special characteristics of participants, or questions about criteria evaluation, a consensus was reached in the study group after discussion. A medical questionnaire part was separated from the assessment part. The medical part, including participant information and written, informed consent, was conducted by study physicians from the Department of Geriatrics. The study physicians were introduced to the study and the execution of the medical questionnaire part by an experienced specialist and senior geriatrician. In case of uncertainties or questions about the evaluation of individual information in the medical part, consultations occurred between the involved physicians and the study team. Results of the medical part were partially available before the mobility assessment was conducted, but they were not disclosed to the physiotherapist. Vice versa, study physicians were not informed about the results of the mobility assessment.

The World Fall Guidelines recommend an algorithm for determining fall risk in seniors [1]. A distinction is made between high risk individuals, individuals with no or low risk, and an intermediate group. According to the World Fall Guidelines, individuals who have not experienced a fall in the previous year and who have none or only a few known fall-promoting factors are categorized as being not at risk. Individuals are considered high risk if they have suffered two or more falls during the past year, needed medical attention due to a fall-related injury, sustained a fracture or other fall-related injury, were unable to get up independently for at least one hour after a fall, and individuals with confirmed frailty and those with loss of consciousness at the event or suspected syncope. For the intermediate fall risk group, not fulfilling low or high-risk criteria, the guidelines recommend conducting walking speed as a standard assessment for further decision-making, with a threshold value of 0.8 meters per second. No separate mobility assessment is recommended for the other risk groups [1].

Walking speed as a gold standard criterion was rejected by the research group, and instead SPPB was chosen. One reason is that the test quality of SIQ was to be captured for all groups of older adults, not just for the group with intermediate risk. In this regard, the guideline recommendation is not suitable for the purpose of our study. Secondly, walking speed is included in the SPPB as a walking part, allowing to make use of walking speed in secondary analyses. Thirdly, clinically different and important information for communication with affected individuals can be derived from the SPPB's three examination parts: standing balance, walking speed, and functional leg strength (5-CRT). Other mobility tests, e.g., walking speed, the Timed Up & Go Test, or others, usually only allow statements about general walking ability or a single functional component of walking, e.g., sole speed. Finally, experience with using the SPPB in our geriatric centre was a further argument for using it.

### e) Additional data collection

Additional data was collected to capture important functional areas and additional risk factors for mobility impairment or falls. Apart from a general assessment of functional abilities by the Longitudinal Urban Cohort Ageing Study Functional

Ability Index (LUCAS FI) [34], we considered the functional areas of mobility, emotion, and cognition, as well as topics such as comorbidities, pain, sarcopenia, and self-assessed quality of life. An overview of the assessment and examination instruments used can be found in S1 Table.

### f) Sample size calculation and statistical analysis

The sample size was calculated according to Hajian-Tilaki [35] for studies on diagnostic tests. It was assumed that the sensitivity of SIQ as a diagnostic test is 75% (Sn = 0.75). High sensitivity is required for tests intended for screening, as tests with high sensitivity are suitable for ruling out healthy or unaffected individuals (28). The lower bound of the 95% confidence interval (95% CI) for sensitivity should not fall below 65% (95% CI ≤ 10%). Type I error was set at 5% (α = 0.05). The prevalence (relative frequency) of an increased fall risk was assumed to be 60% (prev = 0.6) for the entire sample. Under these assumptions and using formula 6.6 from the above-mentioned publication [35], a sample size of 180 individuals was calculated. An increase in the sample size by 5% to 189 individuals was planned to compensate for subsequent exclusion of cases from the analysis, e.g., due to missing values in the main variables (SIQ or SPPB), subsequent identification of exclusion criteria, withdrawal of participants from the study, or other reasons.

The test quality of SIQ as a diagnostic test is presented with typical statistical measures, i.e., sensitivity (Sn), specificity (Sp), positive and negative predictive value (PPV/ NPV), and positive and negative likelihood ratios (LR +, LR-), each with the corresponding 95% CI. The calculation of 95% CI for likelihood ratios is performed using the log method [36], for the other test performance measures using the Wilson method [37], and all other values, e.g., proportions, using normal approximation (Wald method) [38].

Characteristics of the study participants are presented descriptively, for categorical variables with absolute and relative frequencies, for continuous variables with mean and standard deviation. Tests on group differences are conducted, where necessary, with $X^2$ test for categorical and t-test for continuous, symmetrically distributed variables. Adjustment of test results for age and sex is performed using logistic regression, with the SPPB category 'pathological', i.e., SPPB score < 10 points, as the dependent variable, and the explanatory (independent) variables SIQ result, age, and sex. A type I error of ≤ 5% (α ≤ 0.05) is assumed for statistical significance.

The sample size calculation was conducted using spreadsheet software (Microsoft Excel, Office 2019), the calculation of confidence intervals with the software Confidence Interval Analysis (CIA, T. Bryant, University of Southampton, 1994). For all other statistical analyses, SPSS for Windows was used (IBM, SPSS Statistics, Version 27).

### g) Ethics and reporting

Like every scientific investigation involving human beings, this study was conducted in accordance and compliance with the Declaration of Helsinki in its current version [39]. This sub-study was submitted for review to the Ethics Committee of the Hamburg Medical Association as part of the ethics application of the RCT. The Ethics Committee issued a positive vote on June 26, 2023 (registration number 2023–101032-BO-ff). The RCT has been registered in the German Clinical Trials Register (Deutsches Register Klinischer Studien, German Clinical Trials Register DRKS00031408, June 28th, 2023). All participants gave written informed consent.

Our report follows the recommendations for the publication of diagnostic tests of the STARD Initiative (STAndards of Reporting Diagnostic Accuracy Studies) [40]. The STARD checklist is added as S2 Table.

## Results

### a) Recruitment

An overview of the recruitment is shown in Fig 1. The call centre successfully contacted a total of 680 insured individuals residing in Hamburg for the RCT. Contact with 272 individuals occurred outside the recruitment period of this sub-study. All other 408 individuals, regardless of the SIQ results, were offered participation in the sub-study, and about 60%

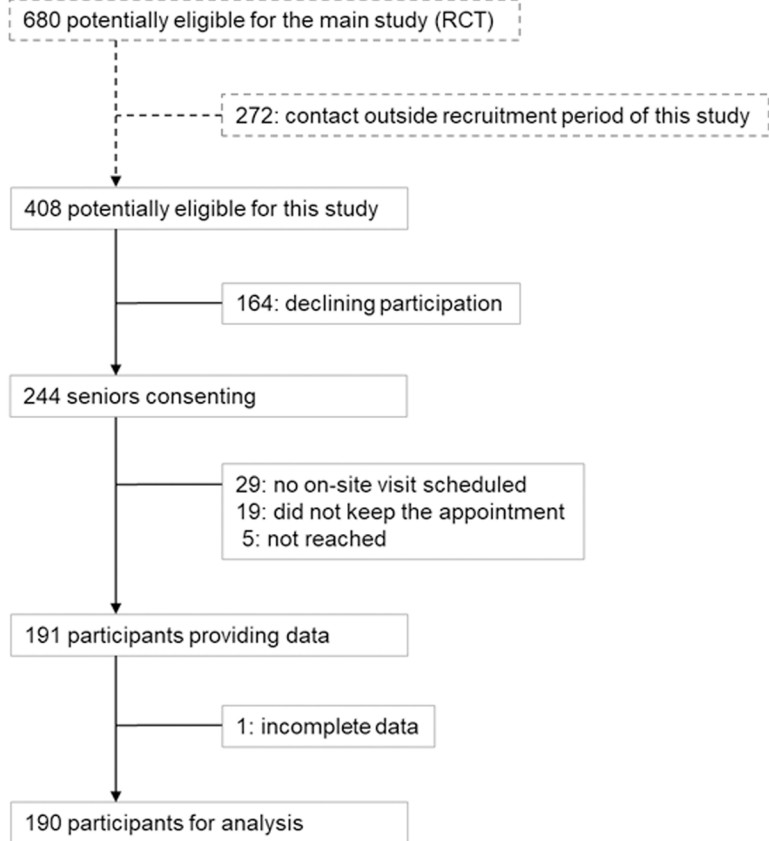

**Fig 1. Study flow-chart.**

of them (244 out of 408 individuals) agreed to participate. Subsequently, about one-fifth of these individuals (53/ 244, 21.7%) did not have an examination appointment: five individuals were unreachable; 19 individuals did not attend the agreed appointment; and with 29 individuals, an appointment arrangement was not successful or an appointment offer was declined. Data from one individual could not be included in the main analysis, because the gold standard SPPB could not be reliably conducted due to blindness of the individual. Thus, data and results from a total of 190 participants are available.

**b) Sample characteristics**

The characteristics of the 190 evaluable study participants are shown in Tables 1 to 3. More women than men have been recruited (143 women, 75.3%). The average age was 80.0 years ± 5.5 years (standard deviation), with women being on average about two years younger than men (79.5 versus 81.5 years, p = 0.04). In the telephone interview, half of the participants reported having fallen in the past year, a fifth reported two or more falls.

An overview of comorbidities captured analogously to the Charlson Comorbidity Index (CCI) is given in Table 2. According to the CCI score, about 30% suffered from advanced comorbidity (CCI ≥ 3 points). The most common diseases covered by the CCI were chronic heart failure (23.7%), chronic lung diseases (21.1%), and diabetes mellitus (16.8%). A positive cancer history was also common in about a quarter of cases. Among other self-reported diseases, high blood pressure (70.0%), osteoarthritis (68.4%), vertigo (60.5%), and lipid metabolism disorders (53.2%) were most frequently

**Table 1. Characteristics of study participants.**

| Characteristic | Frequency (n = 190) | Proportion (%) |
|---|---|---|
| Female sex | 143 | 75.3 |
| **Age (years)** | | |
| ≥ 70 to < 75 | 38 | 20.0 |
| ≥ 75 to < 80 | 54 | 28.4 |
| ≥ 80 to < 85 | 60 | 31.6 |
| ≥ 85 to < 90 | 28 | 14.7 |
| ≥ 90 | 10 | 5.3 |
| Higher educational level (at least ten years of education) | 105 | 55.3 |
| **Body-mass-index (kg/m²)** | | |
| < 22 | 28 | 14.7 |
| ≥ 22 to < 30 | 115 | 60.6 |
| ≥ 30 | 47 | 24.7 |
| **Smoking habits** | | |
| current smoking | 10 | 5.3 |
| former smoking | 109 | 57.4 |
| German care grade | 40 | 21.1 |
| Current use of a walker | 39 | 20.5 |
| **Falls (self-report)** | | |
| at least one fall | 96 | 50.5 |
| frequent falls (≥ 2 falls) | 40 | 21.1 |

Abbreviations: kg: kilogram, m: meter.

mentioned. Sensory impairments (hearing and vision impairment, both around 40%) were also common. Further details can be found in S3 and S4 Tables.

Table 3 summarizes the key results of the geriatric assessment. According to the LUCAS FI, about 20% of the still independently living participants have advanced functional impairment (frail group). Additionally, another 40 individuals (21.1%) have a German care grade. Taken together, more than 40% of individuals were found to have advanced functional impairment. More than four out of ten participants show a slow Timed Up & Go Test (time ≥ 10 sec., 46.8%). In the majority of cases, increased concerns of falling are present. Over half of the participants screened positive for cognitive impairment, about a third for depression.

A comparison of selected characteristics according to falls risk as assessed by SIQ is depicted in Table 4. Overall, participants at risk of falling reported more comorbidities and poorer self-perceived health, and appeared to be more affected by functional impairment in various functional domains. A comparison of single comorbidities can be found in the S4 Table.

### c) Stay Independent Questionnaire as a diagnostic test

According to the translated version of the SIQ, applied by telephone interview, 148 of the 190 participants (77.9%) have been classified as being at increased falls risk. The results according to SIQ were unambiguous, according to the CDC recommendations for its use. The SPPB examination found impairments in standing and walking balance in 81 of 190 participants (42.6%). In the walking component of the SPPB (walking speed), 37 participants (19.5%) were classified as pathological., i.e., with a walking speed < 0.8 meters per second. There were no complications, hazardous situations, or falls during SPPB performance. The SPPB evaluation was also unambiguous according to the existing recommendations for performing and evaluating the SPPB.

**Table 2. Comorbidities analogous to the Charlson Comorbidity Index.**

| Comorbidity | Frequency (n = 190) | Proportion (%) |
|---|---|---|
| **CCI score** | | |
| 0 pts | 62 | 32.6 |
| 1-2 pts | 70 | 36.8 |
| 3-4 pts | 36 | 18.9 |
| ≥ 5 pts | 22 | 11.6 |
| Myocardial infarction | 12 | 6.3 |
| Chronic heart failure | 45 | 23.7 |
| Peripheral arterial disease | 17 | 8.9 |
| Stroke history | 15 | 7.9 |
| Hemiplegia | 4 | 2.1 |
| Dementia | 10 | 5.3 |
| Chronic obstructive lung disease | 40 | 21.1 |
| Rheumatic disease | 14 | 7.4 |
| Peptic ulcer disease | 22 | 11.5 |
| Diabetes mellitus | 32 | 16.8 |
| Moderate to severe kidney failure | 28 | 14.7 |
| Cancer including leukemia | 50 | 26.3 |
| Moderate to severe liver disease | 28 | 14.7 |
| HIV infection | 1 | 0.5 |

Abbreviations: CCI: Charlson Comorbidity Index, pts: points, HIV: Human Immunodeficiency Virus.

The median time span between conducting the telephone interview with SIQ and the examination appointment with SPPB performance was on average 46 days (range: 9–211 days). For six participants, the time span exceeded 90 days. The reported results, particularly regarding the test quality of SIQ, do not change when these six cases are excluded from the analysis.

Fig 2 shows the distribution of the total score of SIQ without special weighting of the question about a fall in the past year. Lower values in SIQ indicate lower fall risk. Fig 3 shows the distribution of the score values of the SPPB. In SPPB, lower score values indicate increasing impairment of standing and walking balance.

In Fig 4, the sample distribution according to the diagnostic test SIQ and the gold standard SPPB is depicted as a two-by-two contingency table. Measured against the gold standard, SIQ correctly identifies 77 individuals as true positive and 38 individuals as true negative. Of 148 SIQ positive individuals, 71 are not impaired in standing and walking balance according to SPPB, i.e., false positive. Of the 42 test-negative individuals, four are actually impaired, i.e., false negative.

Table 5 lists the statistical measures of test performance. Accordingly, SIQ as a diagnostic test has a measured sensitivity of 95.1%, higher than assumed (75%). At the same time, the specificity is low at 34.9%. As assumed for the sample size calculation, the 95% CI for both sensitivity and specificity are < 10%. Corresponding to sensitivity and specificity, a slightly increased positive likelihood ratio (LR+ = 1.46) and a significantly decreased negative likelihood ratio (LR- = 0.14) are found.

The relationship between sensitivity and specificity is illustrated graphically by the receiver operating characteristic (ROC) curve, which is shown in Fig 5a and 5b. In unadjusted analysis (Fig 5a), an area under the curve (AUC) of 65% indicates a rather low to moderate overall performance of SIQ. After adjustment for age and sex, performed with calculated probabilities from a logistic regression model, the test performance improves somewhat, to an AUC of 71% (Fig 5b).

**Table 3. Functional status of study participants.**

| Instrument | Frequency (n = 190) | Proportion (%) |
|---|---|---|
| **LUCAS FI** | | |
| robust | 54 | 28.4 |
| transient | 58 | 30.5 |
| frail | 38 | 20.0 |
| care grade | 40 | 21.1 |
| **Timed Up & Go Test (TUG)** | | |
| < 10 sec | 101 | 53.2 |
| ≥ 10 sec | 89 | 46.8 |
| **Fear of falling (FES-I)** | | |
| low (16–19 pts) | 52 | 27.4 |
| moderate (20–27 pts) | 77 | 40.5 |
| severe (28–64 pts) | 61 | 32.1 |
| **Montreal Cognitive Assessment (MoCA)** | | |
| normal (26–30 pts) | 80 | 42.1 |
| conspicuous (< 26 pts) | 110 | 57.9 |
| **Depressive mood (PHQ-9)** | | |
| normal | 126 | 66.3 |
| mild evidence | 45 | 23.7 |
| moderate or severe evidence | 19 | 10.0 |

Abbreviations: LUCAS FI: Longitudinal Urban Cohort Ageing Study Functional Ability Index, PHQ: Patient Health Questionnaire.

**Table 4. Selected characteristics according to falls risk assessed by the Stay Independent Questionnaire.**

| | SIQ positive | | SIQ negative | | Total | | p-value |
|---|---|---|---|---|---|---|---|
| Characteristic | (n = 148) | | (n = 42) | | (n = 190) | | |
| | n | % | n | % | n | % | |
| Female sex | 107 | 72.3 | 36 | 85.7 | 143 | 75.3 | > 0.05 |
| Age ≥ 80 years | 81 | 54.7 | 17 | 40.5 | 98 | 51.6 | > 0.05 |
| BMI ≥ 30 kg/m² | 40 | 27.0 | 7 | 16.7 | 47 | 24.7 | > 0.05 |
| Smoking (current or former) | 88 | 59.5 | 31 | 73.8 | 119 | 62.6 | > 0.05 |
| Higher education | 46 | 31.1 | 13 | 31.0 | 59 | 31.1 | > 0.05 |
| Low self-perceived health | 87 | 58.8 | 6 | 14.3 | 93 | 49.0 | <0.001 |
| CCI > 2 pt. | 52 | 35.1 | 6 | 14.3 | 58 | 30.5 | < 0.05 |
| German Care grade | 38 | 25.7 | 2 | 4.8 | 40 | 21.1 | 0.001 |
| TUG ≥ 10 sec | 88 | 59.5 | 1 | 2.4 | 89 | 46.8 | <0.001 |
| MoCA < 26 pt. | 89 | 60.1 | 21 | 50.0 | 110 | 57.9 | > 0.05 |
| PHQ-9 category depression | 59 | 39.9 | 5 | 11.9 | 64 | 33.7 | <0.001 |
| SARC-F ≥ 4 pt. | 74 | 50.0 | 2 | 4.8 | 76 | 40.0 | <0.001 |
| FES-I ≥ 14 pt. | 60 | 40.5 | 1 | 2.4 | 61 | 32.1 | <0.001 |
| SPPB < 10 pt. | 77 | 52.0 | 4 | 9.5 | 81 | 42.6 | <0.001 |

Abbreviations: CCI: Charlson Comorbidity Index, FES-I: Falls Efficacy Scale International, kg: kilogram, m: meter, MoCA: Montreal Cognitive Assessment, PHQ: Patient Health Questionnaire, pt: points, SARC-F: Strength, Assistance with walking, Rising from a chair, Climbing stairs, and Falls, sec: seconds, SIQ: Stay Independent Questionnaire, SPPB: Short Physical Performance Battery, TUG: Timed Up & Go Test.

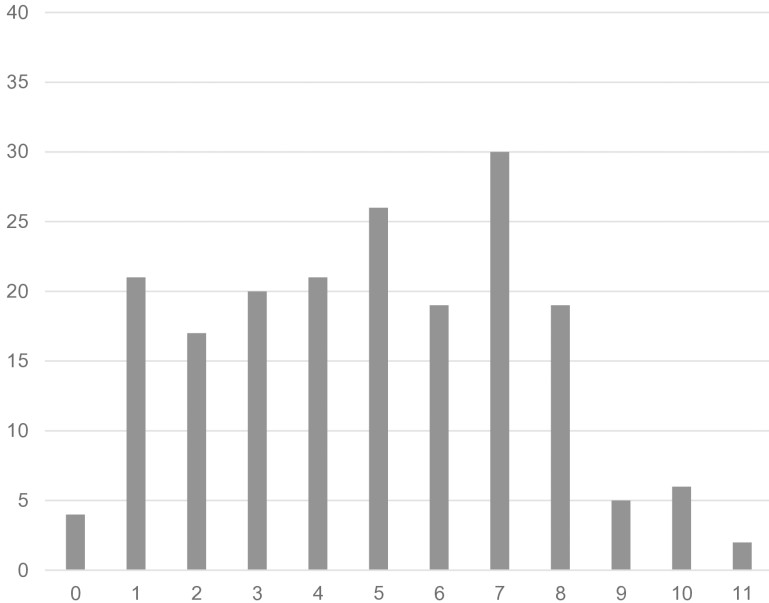

**Fig 2. Distribution of the Stay Independent Qestionnaire (SIQ) total score.** X axis: SIQ total score values (minimum 0 points), with higher values indicating (increasing) risk of falling; Y axis: absolute number of cases. The distribution of the total score of SIQ is depicted without special weighting of the question about a fall in the past year, i.e., with each item couting one point. Lower values in SIQ indicate lower fall risk.

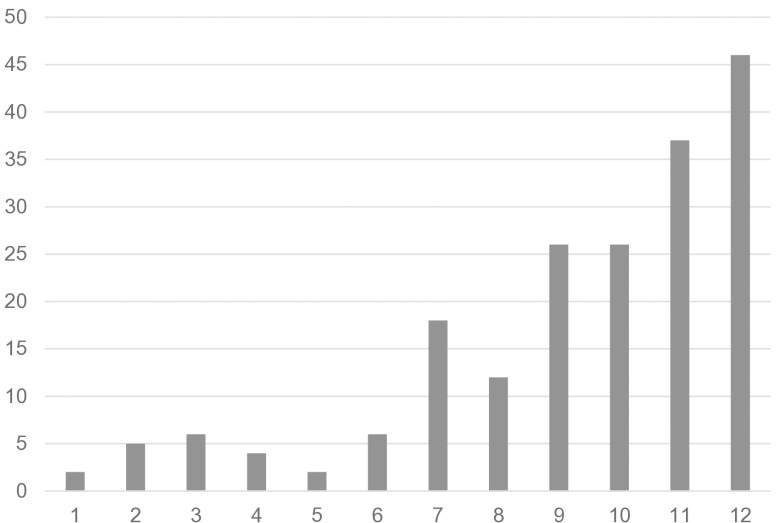

**Fig 3. Distribution of the Short Physical Performance Battery (SPPB) total score.** X axis: SPPB total score values (maximum 12 points), with lower values indicating (increasing) impairments in standing and/or walking balance; Y axis: absolute number of cases.

## Discussion

The project described here has translated the Stay Independent Questionnaire used in the American STEADI prevention program [21] for assessing fall risk in seniors into German and validated it against a standard mobility assessment test, SPPB [41]. This provides another tool for use in German-speaking countries. The advantage of SIQ primarily lies in its

**Fig 4. Contingency table of Stay Independent Questionnaire (SIQ) and Short Physical Performance Battery (SPPB).** SPPB was used as the gold standard, with a total SPPB score < 10 points defining disease (i.e., impairment in standing and/or walking balance); diagnostic test Stay Independent Questionnaire (SIQ), defining risk of falling with a total score of ≥ 4 points.

**Table 5. Performance measures of the Stay Independent Questionnaire (SIQ).**

|  | Performance measure | Estimate | 95% CI |
|---|---|---|---|
| Sn | Sensitivity | 95.1 | [88.0; 98.1} |
| Sp | Specificity | 34.9 | [26.6; 44.2} |
| PPV | Positive predictive value | 52.0 | [44.0; 59.9} |
| NPV | Negative predictive value | 90.5 | [77.9; 96.2} |
| LR+ | Positive likelihood ratio | 1.46 | [1.26; 1.69} |
| LR- | Negative likelihood ratio | 0.14 | [0.05; 0.38} |

Abbreviation: 95% CI: 95% Confidence Interval.

ease of use. It requires answering just twelve relatively straightforward yes/ no questions. The Fall Risk Check, a tool from our research group, covers 13 areas. It requires checking and making between two and seven individual statements per area, which is more cumbersome than SIQ [14]. The Obrist questionnaire even includes 29 predictors for fall risk and has a total of 36 questions [19]. Thus, both previously available German-language tools are more comprehensive and cumbersome than the SIQ questionnaire tested here. In contrast to SIQ, both are less suitable for use in a telephone interview.

The main feature of SIQ, according to the data presented here, is its high sensitivity (~95%). This qualifies SIQ as a simple test that reliably identifies seniors without impairment in standing and walking balance, excluding these individuals from further, presumably unnecessary examinations. The primary purpose of SIQ in the INES project was achieved, namely to exclude insured persons with no or only minor balance impairments and, consequently, presumably no or only minor fall risk from participation in the main study.

A drawback of SIQ is the low specificity measured at about 35%, which means that many individuals without objectively measurable balance impairments are still among those tested positive. Thus, extensive and uncritical screening can lead to unnecessary concerns in individuals with false positive results and increase workload and costs for further diagnostic evaluation. The low specificity may in part be explained by differences in health care systems. The health insurance system established nationwide in Germany is covering almost all medical care situations, which may lead to more and earlier demands for care than in other countries, for example the United States. The SIQ is likely to be helpful when an older person with pronounced concerns of falling consults a primary care physician. In this case, ruling out higher risk with the SIQ may help avoid unnecessary further diagnostics.

Comparing SIQ's test performance with the two known German-language instruments is hardly feasible. For the Fall Risk Check, there is no validation study yet that tests the instrument against a mobility assessment or against the endpoint of falls over time [20]. Obrist et al. validate their online questionnaire against incident falls over six months with monthly follow-up reporting, but do not report sensitivity or specificity or raw data. Only the overall quality from ROC analysis is reported, with an AUC of 0.67 [19]. Comparing the two study populations shows that participants in our study were significantly older (nearly ten years difference) than those in the Obrist study, reported falls more frequently in the past twelve months (about 50% versus about 20%), and generally seem to suffer from significantly more advanced morbidity.

Regarding other instruments, the data presented on SIQ's test performance appear comparable. Most publications report AUC values as an expression of the overall test performance. AUC values between 68% and 73% are described for most tools [15,17,18]. Alessi et al. report a sensitivity of 94% and a specificity of 51% for GPSS, the latter slightly higher but comparable to the values for SIQ [13]. Results for FROP-COM with a sensitivity of 93% and a specificity of 30%, measured at a score threshold of 2 points, are also comparable [15]. A strength of most mentioned studies is that they validate against incident falls during follow-up [13,15,17,18]. While other studies use a twelve-month follow-up period, Hirase's study [17] only follows up for three months, limiting comparability. Unfortunately, it was not possible to conduct a follow-up

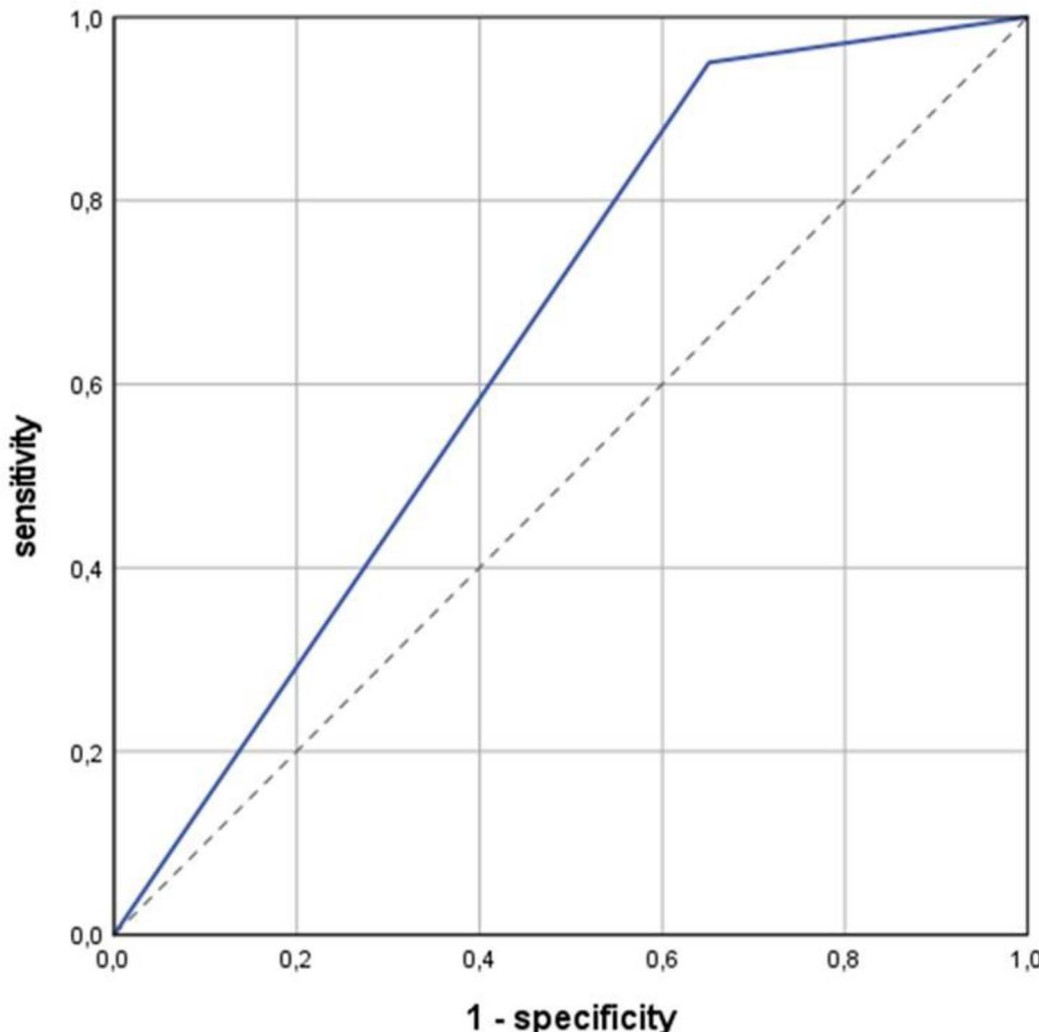

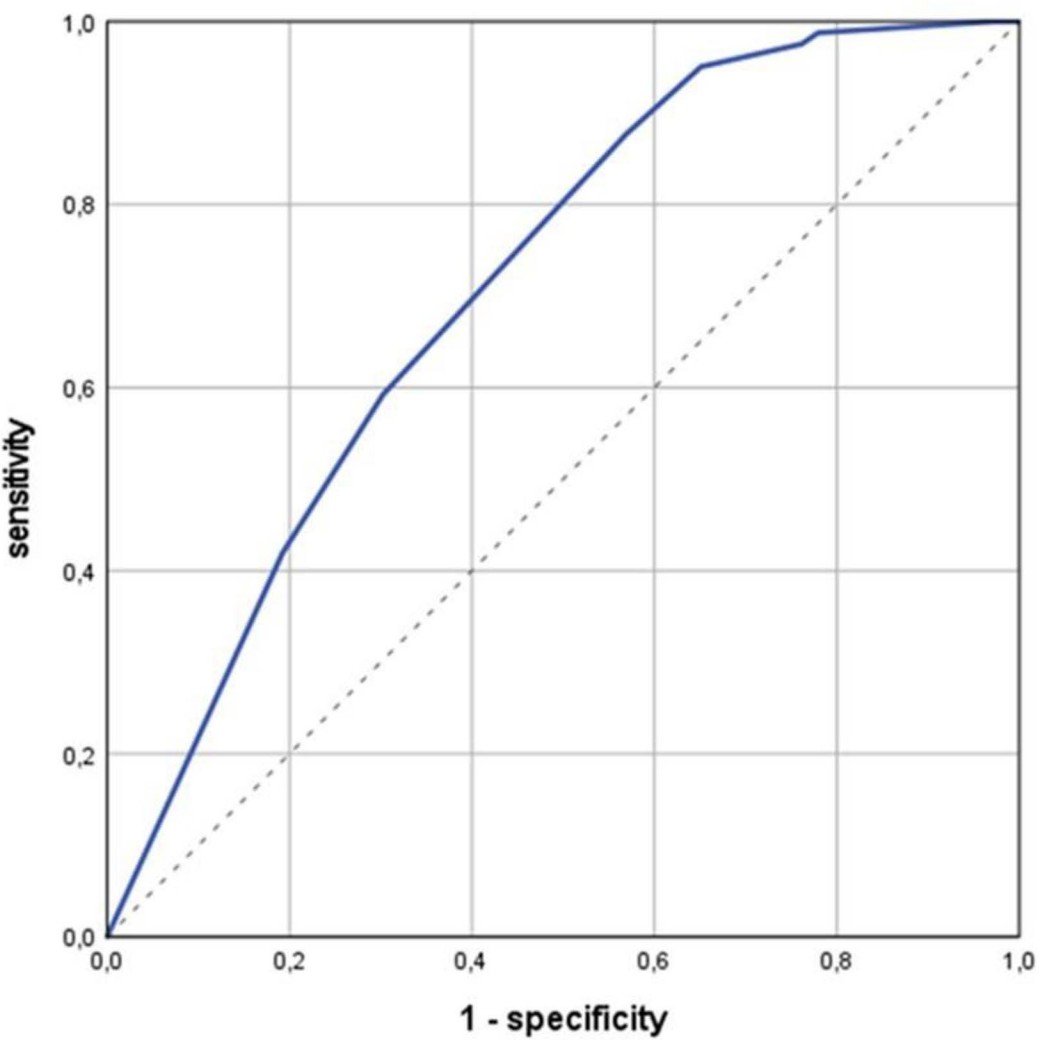

**Fig 5. a. Global test performance of Stay Independent Questionnaire in unadjusted ROC analysis.** Receiver operating characteristic (ROC) curve analysis; target condition: impaired standing and walking balance, as assessed by Short Physical Performance Battery (SPPB score < 10 points); diagnostic test: Stay Independent Questionnaire (SIQ) defined risk of falls (SIQ score ≥ 4 points); blue line: ROC curve; dotted line: reference line; area under the curve (AUC): 0.65 [0.57; 0.73]; unadjusted analysis. 5b. Global test performance of Stay Independent Questionnaire in ROC analysis adjusted for age and sex. Receiver operating characteristic (ROC) curve analysis; target condition: impaired standing and walking balance, as assessed by Short Physical Performance Battery (SPPB score < 10 points); diagnostic test: Stay Independent Questionnaire (SIQ) defined risk of falls (SIQ score ≥ 4 points); blue line: ROC curve; dotted line: reference line; area under the curve (AUC): 0.71 [0.64; 0.78]; adjustment for age and sex.

examination of participants within our project. While incident falls could generally be recorded among participants of the main study, who continue to be followed in the INES project, those testing negative in SIQ, who were excluded from the main study, cannot be questioned and observed any further. We, therefore, opted for a cross-sectional assessment of mobility data and validated the questionnaire against SPPB.

Choosing a gold standard for mobility assessment is still challenging. Current evidence does not clearly indicate which assessment is best suited for detecting mobility impairment or identifying a risk of falling. In the World Fall Guidelines, walking speed, alternatively the Timed Up & Go Test, is preferred over other assessments [1] for the group at intermediate

fall risk. Most recent systematic reviews also recommend walking speed for walking tests, alternatively Timed Up & Go Test, Berg Balance Scale, Performance Oriented Mobility Assessment, or Tandem and Semi-Tandem Stance for balance testing [42–46]. The Chair Rise Test and its modifications are also frequently mentioned [42,44]. As already described, we have chosen SPPB as the gold standard. The recommendation of the World Fall Guidelines is deemed unsuitable for this study's purpose, as we needed a gold standard for all participants, not just for a specific risk group. In communicating about standing and walking ability to test persons, a multi-part assessment like SPPB has advantages over other instruments, which was important for us in the study. Extensive application experience in our study centre has also been an argument for SPPB. Finally, walking speed, Tandem and Semi-Tandem Stance, and Chair Rise Test are included in the three parts of SPPB, so these parameters can be used for further analysis, albeit secondarily.

Pure self-report instruments like SIQ necessarily have limitations. Interestingly, instruments that combine self-reports from test persons, demographic data, and data from assessment or performance tests do not necessarily perform better in terms of test performance criteria. AUC values between 64% and 71% are reported for many instruments and models [47–56]. Data from the modified FROP-COM deviate somewhat, with an AUC of 0.79 [57]. In this tool, age over 80 years, female sex, two or more falls, a tendency toward depression (according to SCL90), hand strength, and postural sway while standing are combined. However, neither this nor any other model has yet established itself as a standard. In a comprehensive systematic review using rigorous criteria for study quality, Gade et al. [58] recently concluded that none of the existing proposals for predicting falls among independently living, community-dwelling seniors is suitable for standard application.

The study presented here has limitations that must be considered when interpreting the results. First, it must be emphasized that SIQ used in STEADI is intended as a self-completion tool and is also used as such. We used the German version of SIQ in a telephone interview. Whether the deviations between self-completion and telephone interview are significant is unclear. However, we cannot quantify such deviation. Therefore, follow-up studies using SIQ as a self-completion tool should follow. Secondly, it must be noted that the preselection of insured persons, at least at the beginning of recruitment, also included coded ICD-10 codes and thus diseases associated with an increased risk of falls. In our view, this does not necessarily affect measures such as sensitivity and specificity, especially since the disease-based preselection has been abandoned in the second half of the recruitment. But certainly, not all characteristics of the participants presented here can automatically be transferred to all seniors living independently. Thirdly, there is evidence that self-selection to some extent took place in our study. As can be seen from the characteristics of the study sample, we recruited much more women than men, and men were significantly older than women, quite the opposite to what is known from the general population. We tried to align at least the AUC measure by performing the ROC analysis with data from a logistic regression model, adjusting for sex and age. However, it is clear that no statistical method is able to correct for selection bias. The question is whether an increased risk of falling may develop with certain comorbidities or special clinical states without leading to a positive test result in the questionnaire – or vice versa. We cannot exclude this, although we consider it unlikely. Further studies are needed to clarify this issue. Finally, it should be mentioned that the sample of insured persons in Hamburg examined in our subproject is not a random sample of the entire INES study population. This means that the characteristics of individuals included in the INES main study as at risk of falling from all three regions may differ from those of the study population we examined in Hamburg.

## Conclusions

The German translation of the SIQ questionnaire is capable of identifying community-dwelling older people without impairment in standing and walking balance, as measured by SPPB.

Thus, SIQ appears to fulfil the main requirement of a screening test for falls risk, i.e., to exclude unimpaired, balance-healthy seniors with presumably low risk.

The recruited group of seniors is characterized by a moderate disease burden, clear functional limitations, and an at least moderately increased fall risk. This might also be expected for the entire study population of the ongoing RCT.

## Supporting information

**S1 Table. Assessment instruments and questionnaires for data collection.**
(DOCX)

**S2 Table. STARD checklist.**
(DOCX)

**S3 Table. Further comorbidities apart from comorbidities of the Charlson Comorbidity Index.**
(DOCX)

**S4 Table. Comorbidities according to falls risk assessed by the Stay Independent Questionnaire (SIQ).**
(DOCX)

## Acknowledgments

We gratefully acknowledge Ms. S. Stasch for excellent secretarial work and support, and Ms. B. Ruß-Thiel and Mr. T. Schaknat, ife Gesundheits-GmbH, Nehmten, Germany, for their cooperation and for organising and performing the telephone interviews. Our sincere thank goes to all study participants and their relatives interested in our work and supporting this study.

## Author contributions

**Conceptualization:** Ulrich Thiem, Ulrike Dapp, Saskia Otte.

**Data curation:** Stefan Golgert.

**Formal analysis:** Ulrich Thiem, Stefan Golgert.

**Funding acquisition:** Ulrich Thiem, Ulrike Dapp.

**Investigation:** Ingeborg Schlumbohm, Saskia Otte.

**Methodology:** Ulrich Thiem, Stefan Golgert, Ulrike Dapp.

**Project administration:** Ulrich Thiem, Ingeborg Schlumbohm, Ulrike Dapp, Saskia Otte.

**Resources:** Ulrich Thiem, Saskia Otte.

**Software:** Stefan Golgert.

**Supervision:** Ulrich Thiem.

**Writing – original draft:** Ulrich Thiem.

**Writing – review & editing:** Ingeborg Schlumbohm, Stefan Golgert, Ulrike Dapp, Saskia Otte.

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
