## [Decision Letter · Decision Letter 0]

30 Apr 2025

Dear Dr. Thiem,

Thank you for submitting your manuscript to PLOS ONE. After careful consideration, we feel that it has merit but does not fully meet PLOS ONE’s publication criteria as it currently stands. Therefore, we invite you to submit a revised version of the manuscript that addresses the points raised during the review process.

Kindly review the comments provided by the reviewers below and revise the manuscript as appropriate to reflect their feedback.

We look forward to receiving your revised manuscript.

Kind regards,

Hansani Madushika Abeywickrama, Ph.D.

Academic Editor

PLOS ONE

Journal Requirements:

Reviewers' comments:

Reviewer's Responses to Questions

**Comments to the Author**

1. Is the manuscript technically sound, and do the data support the conclusions?

Reviewer #1: Yes

Reviewer #2: Yes

Reviewer #3: Yes

Reviewer #4: Yes

2. Has the statistical analysis been performed appropriately and rigorously?

Reviewer #1: Yes

Reviewer #2: Yes

Reviewer #3: Yes

Reviewer #4: Yes

3. Have the authors made all data underlying the findings in their manuscript fully available?

Reviewer #1: Yes

Reviewer #2: Yes

Reviewer #3: No

Reviewer #4: Yes

4. Is the manuscript presented in an intelligible fashion and written in standard English?

Reviewer #1: Yes

Reviewer #2: Yes

Reviewer #3: Yes

Reviewer #4: Yes

Reviewer #1: Thank you for your efforts, please consider:

1- Conducting a proofreading for the manuscript

2- Explain the rationale of the study

3- Highlight the importance and impact of study at micro and macro-levels

4- Shed light on the population dynamic in Germany with special focus on aging population

5- Have you published fall risk check and if not, what was the validation process for the tool

6- Explain the role of the expert committee

7- Detail validity assessment

8- Detail reliability assessment

9- Detail the pilot preliminary testing

Reviewer #2: The manuscript presents nicely the validity of a German version applied by phone of the Stay Independent Questionnaire, a validated tool to assess the risk of falls in older adults. The results are important as they showed that although the specificity is not great, this telephonic tool has a high sensitivity, which is relevant to prevent falls in older adults who are more at risk of having severe consequences. There are some comments to consider that I’m outlining below:

I’m not sure if the SIQ questionnaire is originally a telephonic test, or if it was adapted

Line 116: states “The mentioned instruments vary considerably.” I’m not sure what vary considerably.

Line 116-121: states that “The Fall Risk Check, the only German questionnaire”, but later states that there is another questionnaire in German (the Swiss). Why the group did not use this questionnaire? Please clarify

Needs to develop more the rationale behind translating the SIQ, it is not very clear why this questionnaire is a good tool compared to other instruments (it’s only stated at lines 138-139).

Line 146: hypothesis section is confusing; it claims that there’s no statistical hypothesis but then a hypothesis is formulated. I suggest to 1) remove the section, 2) work on a better way to define the hypothesis (there were statistical analysis done so there should be a hypothesis to test).

Line 530: Is there also any possibility that some people who self-excluded from the study may have higher risks or had more falls in the past, or be reluctant to have medical appointments? Maybe discuss this point if there is a possibility that this fact may have influenced the sample.

Reviewer #3: The study by Thiem et al. (2024) aimed to validate the German version of the Stay Independent Questionnaire (SIQ) for identifying community-dwelling seniors aged 70 and older at risk of falling. The study is well-structured, providing a clear rationale, methodology, results, and discussion.

Address the high false positive rate further. Expand the discussion to thoroughly address the implications of the high false positive rate. Specifically, discuss the potential consequences of a high false positive rate in clinical practice (e.g., increased healthcare costs due to unnecessary referrals, patient anxiety). Explore potential reasons for the low specificity in the German context. Could cultural factors, language nuances, or differences in healthcare systems play a role? Suggest strategies to mitigate the high false positive rate, such as combining the SIQ with other screening tools or using a higher cut-off score (although they did explore this).

Reviewer #4: The study is well explained though out the documents but recommend to write rationale and aim in very clear format. Please explain translators and translation process.

Validation method is mentioned but more information regarding the agreement of the new items and if any modification were done during translation .

**Do you want your identity to be public for this peer review?** For information about this choice, including consent withdrawal, please see our Privacy Policy

Reviewer #1: **Yes: ** WEAM BANJAR

Reviewer #2: No

Reviewer #3: No

Reviewer #4: **Yes: ** Ahmed Ibrahim Al Kharusi

---

## [Author Response · Author response to Decision Letter 1]

12 Jul 2025

Response to the Reviewers

Reviewer #1: Thank you for your efforts, please consider:

1- Conducting a proofreading for the manuscript

Done. Examples:

Abstract, line 80: “… seniors aged 70 and older without impairment in standing and walking balance” instead of “… seniors aged 70 and older with impaired standing and walking balance”.

Background, lines 97-99: “Repeated falls can lead to a diminished quality of life due to pain and injuries.” instead of “Repeated falls lead to impaired quality of life through pain and injuries.”

Background, lines 117-121: “As a low-threshold approach to identifying community-dwelling seniors at increased risk of falling, various self-report questionnaires have been developed both nationally and internationally.” instead of “As a low-threshold approach to identify community-dwelling seniors at increased risk of falling, various self-report questionnaires have been developed nationally and internationally.”

Background, lines 131-134: “For most tools, evaluation has been limited, with often only a single publication describing the development of the instrument.” instead of “For most tools, the extent of evaluation is limited. Often, only one publication exists that describes the development of the instrument”.

Background, line 158-161: “We chose SIQ and decided to translate it into German and validate it against a standard mobility assessment.” instead of “Our choice fell on SIQ, and we decided to translate it into German and validate it against a standard mobility assessment.”

Methods, line 171: “…without impairment in standing and walking balance” instead of “… with unimpaired standing and walking balance”.

Methods, line 172: “… was assumed […] with a sensitivity of at least 75%.” instead of “…was assumed […] demonstrating a sensitivity of at least 75%.”

Methods, line 208-210: “About a third of the participants claimed reimbursement for travel or taxi costs.” instead of “About a third of the participants took advantage of this by getting travel or taxi costs reimbursed.”

Methods, line 255-258: “As the validation study took place later, the call centre had no participant information from the subproject.” instead of “Since the investigation in this validation study occurred subsequently, there was no information from the subproject about participants at the call centre.”

Methods, line 262-266: “Because of the health insurance companies' eligibility criteria, the call centre recruited more seniors with fall risk than without when contacting potential participants for the main study.” instead of “Due to the eligibility criteria of the health insurance companies when contacting potential participants for the main study, overall, more seniors with fall risk than without were found via the recruitment of the call centre.”

Results, lines 427-429: “Over half of the participants screened positive for cognitive impairment, about a third for depression.” instead of “Over half of the participants are conspicuous in screening for cognitive impairment, about a third in depression screening.”

Results, lines 432-434: “Overall, participants at risk of falling reported more comorbidities and poorer self-perceived health…” instead of “Overall, participants at falls risk reported more comorbidities and worser self-perceived health…”

Results, lines 445-447: “In the walking component of the SPPB (walking speed), 37 participants (19.5%) were classified as pathological…” instead of “In the walking part of the SPPB alone, with measurement of walking speed, 37 participants (19.5%) were pathological…”.

Discussion, lines 625-629: “This means that the characteristics of individuals included in the INES main study as at risk of falling from all three regions may differ from those of the study population we examined in Hamburg.” instead of “This means that the characteristics of those individuals included in the INES main study as at-risk of falling from all three regions may well differ from the characteristics of the study population in Hamburg that we examined.”

2- Explain the rationale of the study

We extended our explanation of the study rationale, starting in line 147.

3- Highlight the importance and impact of study at micro and macro-levels

We added two sentences in the introduction on recent fracture incidences for Germany and the aging population (lines 104-106), and also added a discussion of the disadvantage of the single question approach as recommended by the World Falls Guideline (line 147-154).

4- Shed light on the population dynamic in Germany with special focus on aging population

Done, see above.

5- Have you published fall risk check and if not, what was the validation process for the tool

The development of the Fall Risk Check and first data for evaluation have been published by Anders J et al. (2006) and Anders J et al. (2007), referenced as [14] and [20]. As mentioned in the discussion section (line 548-550), the Fall Risk Check has not yet been validated against standard mobility assessment.

6- Explain the role of the expert committee

The expert committee reviewed the translations of the Stay Independent Questionnaire, discussed them with a bilingual scientist experienced in geriatrics, and compiled the final version of the German translation. The committee also discussed the results and the feedback of volunteers of the pilot testing.

7- Detail validity assessment

The validation of the Stay Independent Questionnaire (SIQ) was performed by administering the questionnaire and performing the Short Physical Performance Battery in all participants. Details can be found in the methods section, part c) German Translation of the Stay Independent Questionnaire and d) Test administration. The questionnaire was not investigated in terms of construct validity and the like. Rather, the aim was to take the instrument - or better the German translation of it - as it is and test it against a standard mobility assessment. We agree with the reviewer that investigating the questionnaire further is valuable. However, this is beyond the intention of the current manuscript.

8- Detail reliability assessment

Reliability was not assessed in this study.

9- Detail the pilot preliminary testing

After short instruction by staff personnel, eight patients of the day clinic associated to the Department of Geriatrics voluntarily answered the self-administered German SIQ version. After completion, issues like clarity of the questionnaire and single items, wording etc. were discussed with the volunteers. Feedback was received, but did not lead to further alterations of the questionnaire.

This information has been added to the text, see lines 232-236.

Thank you for all comments and suggestions.

Reviewer #2: The manuscript presents nicely the validity of a German version applied by phone of the Stay Independent Questionnaire, a validated tool to assess the risk of falls in older adults. The results are important as they showed that although the specificity is not great, this telephonic tool has a high sensitivity, which is relevant to prevent falls in older adults who are more at risk of having severe consequences.

Thank you.

There are some comments to consider that I’m outlining below:

1. I’m not sure if the SIQ questionnaire is originally a telephonic test, or if it was adapted

The original SIQ is a self-administered questionnaire. However, the questions are easy to administer without any adaption in a telephone interview, as we did.

Line 116: states “The mentioned instruments vary considerably.” I’m not sure what vary considerably.

We changed the sentence into: “The mentioned instruments vary considerably with regard to included items, length and ease of use.” see lines 125-126

Line 116-121: states that “The Fall Risk Check, the only German questionnaire”, but later states that there is another questionnaire in German (the Swiss). Why the group did not use this questionnaire? Please clarify

Sorry for being confusing here. The first questionnaire is the only one from Germany, but a second questionnaire in German language exists, coming from Switzerland.

We changed this in the text: “The Fall Risk Check, one of two German questionnaires…” (line 126), and “The online questionnaire by Obrist, the other German-language tool…” (line 128).

The Swiss questionnaire was not used, because it is much longer than the SIQ.

Needs to develop more the rationale behind translating the SIQ, it is not very clear why this questionnaire is a good tool compared to other instruments (it’s only stated at lines 138-139).

The Fall Risk Check covers 13 areas with in total 50 statements to be checked, the Obrist questionnaire includes 29 predictors of fall risk with a total of 36 questions. SIQ is shorter and appeared to be easier in use, and hence we decided to use it.

We added this to the introduction (see line 127 and lines 158-161).

Line 146: hypothesis section is confusing; it claims that there’s no statistical hypothesis but then a hypothesis is formulated. I suggest to 1) remove the section, 2) work on a better way to define the hypothesis (there were statistical analysis done so there should be a hypothesis to test).

Sorry for being confusing here.

Our interpretation of a statistical hypothesis is that it can be tested by a significance / inference test, like Χ² test etc. This is not appropriate for diagnostic tests and measures like sensitivity, specificity and the like. There is no comparison of two or more groups in a normal diagnostic accuracy study, and hence no inference test. That was meant with our statement that a testable hypothesis was not formulated.

To avoid confusion, we simply removed the sentence: “A study hypothesis to be statistically tested was not formulated.” see line 169

Line 530: Is there also any possibility that some people who self-excluded from the study may have higher risks or had more falls in the past, or be reluctant to have medical appointments? Maybe discuss this point if there is a possibility that this fact may have influenced the sample.

Thank you very much, this is an important point, addressing possible selection bias due to self-selection.

We have evidence that self-selection to some extent took place in our study. As can be seen from the characteristics of the study sample, we recruited much more women than men, and men were significantly older than women – quite the opposite to what is known from the general population. We tried to align at least the AUC measure by performing the ROC analysis with data from a logistic regression model, adjusting for sex and age. However, it is clear that no statistical method is able to correct for selection bias.

The question is whether an increased risk of falling may develop with certain comorbidities or special clinical states without leading to a positive test result in the questionnaire – or vice versa. We cannot imagine this, although this is only speculation.

We added the point of selection bias as a limitation to the discussion section (see last paragraph in the discussion, lines 614-623).

Reviewer #3: The study by Thiem et al. (2024) aimed to validate the German version of the Stay Independent Questionnaire (SIQ) for identifying community-dwelling seniors aged 70 and older at risk of falling. The study is well-structured, providing a clear rationale, methodology, results, and discussion.

Thank you.

Address the high false positive rate further. Expand the discussion to thoroughly address the implications of the high false positive rate. Specifically, discuss the potential consequences of a high false positive rate in clinical practice (e.g., increased healthcare costs due to unnecessary referrals, patient anxiety).

Thank you very much for this important comment. Indeed, the high sensitivity alone does not qualify the questionnaire as an ‘ideal’ screening instrument. For this, specificity needed to be much higher.

As requested, we added the following sentences to the discussion section (lines 539-546):

“Thus, extensive and uncritical screening can lead to unnecessary concerns in individuals with false positive results and increase workload and costs for further diagnostic evaluation. The low specificity may in part be explained by differences in health care systems. The health insurance system established nationwide in Germany is covering almost all medical care situations, which may lead to more and earlier demands for care than in other countries, for example the United States. The SIQ is likely to be helpful when an older person with pronounced concerns of falling consults a primary care physician. In this case, ruling out higher risk with the SIQ may help avoid unnecessary further diagnostics.”

Explore potential reasons for the low specificity in the German context. Could cultural factors, language nuances, or differences in healthcare systems play a role?

One may speculate that differences between health care systems may contribute to the low specificity. We can imagine that the health insurance system in Germany established nationwide and covering almost all care situations may lead to earlier demands for care in Germany than in other countries, especially the United States.

We added this point to the discussion section (see above).

Suggest strategies to mitigate the high false positive rate, such as combining the SIQ with other screening tools or using a higher cut-off score (although they did explore this).

As already pointed out in the discussion, instruments that combine self-reported information with data from assessment or performance tests do not necessarily perform better in predicting mobility impairment (see discussion section, lines 556-566). Hence, this would be an argument against combining the SIQ with performance measures. What we can offer is to use SIQ in case people contact their primary care physician to express high concerns of falling. In this case, the SIQ – easy to administer and low at costs – can be offered to identify people at low risk for whom further work-up can be suspended.

We added this point to the discussion section (see above).

Reviewer #4:

The study is well explained though out the documents but recommend to write rationale and aim in very clear format.

We rephrased the paragraphs dealing with the study rationale (see changes in the introduction, and also response to reviewer #1).

Please explain translators and translation process.

We rephrased the paragraph on the translation process:

“The CDC's questionnaire version was translated into German by one and back-translated by another professional translator, with the second translator being blinded to the original version. The back-translation and the original version were compared, and the translation was finalised by the study group with the help of a bilingual scientist, experienced in geriatric medicine. The German questionnaire was then pre-tested with eight seniors visiting the geriatric day clinic associated to the Department of Geriatrics. After short instruction by staff personnel, the patients voluntarily answered the self-administered German SIQ version. After completion, issues like clarity of the questionnaire, of single items, wording etc. were discussed with the volunteers. Feedback was received, but did not lead to further alterations of the questionnaire.”, see lines 227-236.

Validation method is mentioned but more information regarding the agreement of the new items and if any modification were done during translation .

As already mentioned (see general issues and response to reviewer #1, point 7) the aim of our study has been to assess the validity of der Stay Independent Questionnaire in a clinical context. The questionnaire was not investigated in terms of construct validity and the like. Rather, the aim was to take the instrument - or better the German translation of it - as it is and test it against a standard mobility assessment. With this, we adhere to all recommendations of the STARD criteria for validating diagnostic tests.

Thank you for critically reading and commenting our manuscript.

---

## [Decision Letter · Decision Letter 1]

5 Aug 2025

Validity of the German version of the Stay Independent Questionnaire applied by telephone interview: A diagnostic accuracy study

PONE-D-25-04785R1

Dear Dr. Thiem,

We’re pleased to inform you that your manuscript has been judged scientifically suitable for publication and will be formally accepted for publication once it meets all outstanding technical requirements.

Kind regards,

Hansani Madushika Abeywickrama, Ph.D.

Academic Editor

PLOS ONE

Additional Editor Comments (optional):

Reviewers' comments:

Reviewer's Responses to Questions

**Comments to the Author**

Reviewer #2: All comments have been addressed

Reviewer #3: All comments have been addressed

2. Is the manuscript technically sound, and do the data support the conclusions?

Reviewer #2: Yes

Reviewer #3: Yes

3. Has the statistical analysis been performed appropriately and rigorously?

Reviewer #2: Yes

Reviewer #3: Yes

4. Have the authors made all data underlying the findings in their manuscript fully available?

Reviewer #2: Yes

Reviewer #3: No

5. Is the manuscript presented in an intelligible fashion and written in standard English?

Reviewer #2: Yes

Reviewer #3: Yes

Reviewer #2: I think the authors have done a good job addressing reviewer's comments. A minor comment is that I would prefer the sentence "A study hypothesis to be statistically tested was not formulated." in line 148 removed (just a personal preference).

Reviewer #3: (No Response)

**Do you want your identity to be public for this peer review?** For information about this choice, including consent withdrawal, please see our Privacy Policy

Reviewer #2: No

Reviewer #3: No

---

## [Editor Report · Acceptance letter]

PONE-D-25-04785R1

PLOS ONE

Dear Dr. Thiem,

I'm pleased to inform you that your manuscript has been deemed suitable for publication in PLOS ONE. Congratulations! Your manuscript is now being handed over to our production team.

Kind regards,

on behalf of

Dr. Hansani Madushika Abeywickrama

Academic Editor

PLOS ONE